# Redox Balance in Male Infertility: Excellence through Moderation—“Μέτρον ἄριστον”

**DOI:** 10.3390/antiox10101534

**Published:** 2021-09-27

**Authors:** Evangelos N. Symeonidis, Evangelini Evgeni, Vasileios Palapelas, Dimitra Koumasi, Nikolaos Pyrgidis, Ioannis Sokolakis, Georgios Hatzichristodoulou, Chara Tsiampali, Ioannis Mykoniatis, Athanasios Zachariou, Nikolaos Sofikitis, Ares Kaltsas, Fotios Dimitriadis

**Affiliations:** 1Department of Urology, “G. Gennimatas” General Hospital, Aristotle University of Thessaloniki, 54635 Thessaloniki, Greece; evansimeonidis@gmail.com (E.N.S.); g_mikoniatis@hotmail.com (I.M.); 2Cryogonia Cryopreservation Bank, 11526 Athens, Greece; lina.evgeni@cryogonia.gr (E.E.); cryolab@cryogonia.gr (D.K.); 33rd Department of Obstetrics and Gynecology, Hippokration General Hospital, School of Medicine, Aristotle University, 54642 Thessaloniki, Greece; palapelas@the.forthnet.gr; 4Department of Urology, ‘Martha-Maria’ Hospital Nuremberg, 90491 Nuremberg, Germany; nikospyrgidis@gmail.com (N.P.); sokolakisi@gmail.com (I.S.); drhatzichristodoulou@gmail.com (G.H.); 5Independent Researcher, 54250 Thessaloniki, Greece; x.tsiampali@gmail.com; 6Department of Urology, School of Medicine, Ioannina University, 45500 Ioannina, Greece; zahariou@otenet.gr (A.Z.); v.sofikitis@hotmail.com (N.S.); ares-kaltsas@hotmail.com (A.K.)

**Keywords:** male infertility, redox balance, reactive oxygen species, oxidative stress, reductive stress, antioxidants, antioxidant capacity

## Abstract

Male infertility, a relatively common and multifactorial medical condition, affects approximately 15% of couples globally. Based on WHO estimates, a staggering 190 million people struggle with this health condition, and male factor is the sole or contributing factor in roughly 20–50% of these cases. Nowadays, urologists are confronted with a wide spectrum of conditions ranging from the typical infertile male to more complex cases of either unexplained or idiopathic male infertility, requiring a specific patient-tailored diagnostic approach and management. Strikingly enough, no identifiable cause in routine workup can be found in 30% to 50% of infertile males. The medical term male oxidative stress infertility (MOSI) was recently coined to describe infertile men with abnormal sperm parameters and oxidative stress (OS), including those previously classified as having idiopathic infertility. OS is a critical component of male infertility, entailing an imbalance between reactive oxygen species (ROS) and antioxidants. ROS abundance has been implicated in sperm abnormalities, while the exact impact on fertilization and pregnancy has long been a subject of considerable debate. In an attempt to counteract the deleterious effects of OS, urologists resorted to antioxidant supplementation. Mounting evidence indicates that indiscriminate consumption of antioxidants has led in some cases to sperm cell damage through a reductive-stress-induced state. The “antioxidant paradox”, one of the biggest andrological challenges, remains a lurking danger that needs to be carefully avoided and thoroughly investigated. For that reason, oxidation-reduction potential (ORP) emerged as a viable ancillary tool to basic semen analysis, measuring the overall balance between oxidants and antioxidants (reductants). A novel biomarker, the Male infertility Oxidative System (MiOXSYS^®^), is a paradigm shift towards that goal, offering a quantification of OS via a quick, reliable, and reproducible measurement of the ORP. Moderation or “*Μέτρον*” according to the ancient Greeks is the key to successfully safeguarding redox balance, with MiOXSYS^®^ earnestly claiming its position as a guarantor of homeostasis in the intracellular redox milieu. In the present paper, we aim to offer a narrative summary of evidence relevant to redox regulation in male reproduction, analyze the impact of OS and reductive stress on sperm function, and shed light on the “antioxidant paradox” phenomenon. Finally, we examine the most up-to-date scientific literature regarding ORP and its measurement by the recently developed MiOXSYS^®^ assay.

## 1. Introduction

Infertility, the incapacity to conceive after one year of regular, unprotected sexual intercourse, affects approximately one out of five of couples worldwide [1,2,3]. Male factor contributes to almost half of all cases, and roughly 7% of men are afflicted on a global scale [4,5,6]. Male infertility is multifactorial and oxidative stress (OS) is at the epicenter of pathogenesis, as shown in both human and animal studies, with a stunning percentage of OS-related cases ranging from 30% to 80% in selected studies [4,7,8,9,10].

Strikingly enough, no specific cause can be identified in 30% to 50% of infertile males in the absence of female factor infertility [7]. These patients are characterized as having idiopathic infertility, a challenging condition, with overwhelming evidence pointing towards OS’s critical role [7,11]. OS, one of the leading causes of male infertility, delineates an imbalance between oxidants and reductants, mainly deriving from either surplus production of reactive oxygen species (ROS) or a deficiency of antioxidants [6,12]. It is a condition that has been extensively studied in male infertility, and various exogenous or endogenous sources have already been recognized as leading causes of increased ROS production. Varicocele, leukocytes, immature spermatozoa, smoking, alcohol, radiation, and toxins are among the most prominent [13].

Solid evidence also demonstrates that OS exerts deleterious effects on healthy spermatozoa via these oxygen-derived free radicals [13,14]. Spermatozoa are susceptible to ROS-induced damage owing to their structural pattern. Their plasma membrane consists of polyunsaturated fatty acids, making their composition considerably vulnerable to lipid peroxidation [4]. The detrimental effects of ROS expand equally to mitochondrial and nuclear DNA and sperm epigenome [5,6]. Chromosomal microdeletions, increased DNA fragmentation, and mutations are integral components of DNA damage [6]. Indeed, mitochondria are not only the target of ROS, but also a significant source, thus contributing to a continuous and prolonged cycle of ROS production [5].

From a clinical standpoint, OS translates into declined fertilization rates, defective embryonic development, recurrent miscarriage, and overall inferior assisted reproductive technology (ART) outcomes [5]. Nonetheless, it should be underlined that a physiological amount of ROS is always required for the critical processes of capacitation, hyperactivation, acrosome reaction, and sperm–oocyte binding [5,15].

Agarwal et al. recently proposed the terminology of male oxidative stress infertility (MOSI), targeting a better evaluation of infertile patients with coexistent OS. Besides, this classification incorporated males previously considered to suffer from idiopathic infertility [7].

Hence, it becomes even more important that OS cannot be overlooked, and male infertility should not be underserved. The need to test for OS is of great importance and could provide avenues in assessing a male’s reproductive potential adequately while ensuring a better prognosis [16]. In this context, the diagnostic workup of the infertile male seems to go far beyond the conventional semen analysis [17]. Although broadly used and still the reference diagnostic standard, it is hindered by several inherent shortcomings. Its reliability has been constantly judged in terms of both technical and biological reasons. High intra-individual variability and inconsistency in interpreting functional sperm properties at a molecular level pose a diagnostic challenge [18]. Therefore, the necessity for advanced sperm function tests, which will accurately measure OS and assist in the diagnosis of male infertility, has justifiably emerged [7].

Currently available diagnostic tools that can measure the levels of OS are characterized by innate imperfections impeding their wide adoption in routine clinical practice. For that reason, their use has undergone rampant criticism, giving space for novel in vitro diagnostic methods to unfold [7,19]. Male infertility Oxidative System (MiOXSYS^®^) confers many advantages, thereby offering a rapid and reliable in vitro analysis of semen’s oxidative stress [20]. It is unquestionably a new diagnostic hallmark for precise OS detection and generally a practical and easy-to-use method for oxidation-reduction potential (ORP) quantification [20].

For years, urologists have aimed at tackling the OS with the quick and relatively safe option of antioxidants [21]. These widely used regimens appeared viable and inexpensive solutions against OS with demonstrable effectiveness in improving sperm parameters and live birth rates [7]. Nowadays, there is growing awareness that uncontrolled therapy is not devoid of risks. More specifically, such an approach is deemed risky, and there is always the danger of diversion to a reductive stress state [22]. The antioxidant paradox, virtually a paradoxical effect, is a terminology that has been formulated to describe the detrimental side effects from the indiscriminate administration of these compounds [19,22].

Our article’s title, “*Μέτρον ἄριστον/Metron ariston*” (excellence through moderation), stems from the aphorism “*Excellence through moderation*”, implying that doing or having too much of one thing is not always beneficial. On the contrary, it may be somehow detrimental, even if that thing is generally considered beneficial. Apart from excellence, “*ἄριστον*” was also considered a quick lunch taken around noon or early afternoon in ancient times, a fact that parallels the antioxidant administration. Hence, we sought to convey the simultaneous message of both excellence and nutrition.

Τhe above-mentioned quote is attributed to the Greek poet, and one of the seven sages of ancient times, Cleobulus (*Greek: Κλεόβουλος ὁ Λίνδιος, Kleoboulos ho Lindios*). Historically, the seven wise men (*Greek: οἱ ἑπτὰ σοφοί hoi hepta sophoi*) were philosophers, statesmen, and law-givers who lived around the sixth century B.C. and were renowned for their wisdom.

Ancient Greeks strongly believed in a life full of harmony and far from extremities. The significance of “*Μέτρον ἄριστον*” lies in the fact that every aspect of our lives should be strictly characterized by mean. In other words, it could be virtually advocated that “best” is what is used reasonably. Transferring the concept of moderation to male infertility, urologists need to embark upon continuous endeavors towards a subtle balance in redox state [1,6].

This review summarizes all the existing evidence relevant to redox balance in male infertility, thereby highlighting the importance of moderation in every aspect of reproduction. Furthermore, we discuss the role of oxidative and reductive stress in the field, critically analyze the “antioxidant paradox”, and uncover valuable insights into the use of antioxidants. Finally, we describe the oxidation-reduction potential (ORP), the latest diagnostic advancement in male infertility, and its measurement with the promising MiOXSYS^®^ assay.

## 2. Search Strategy

A comprehensive non-systematic electronic search was performed in July 2021 within the Medline via PubMed interface with the following interchangeable combination of keywords: “redox balance”, “redox imbalance”, “redox regulation”, “male subfertility”, “male infertility”, “oxidative stress”, and “reductive stress”. Boolean operators (AND, OR) were also applied in succession to form our search strategy better. We aimed to primarily identify all relevant articles to redox balance in male infertility.

We included only articles written in the English language with no specific date restrictions. Reviews, meta-analyses, prospective and retrospective original research studies, editorials, research letters, letters to the editor, brief communications, opinion papers, and commentaries were examined and thoroughly analyzed based on their clinical relevance. Studies reporting on irrelevant topics, studies with insufficient data, interventions not clearly reported as an antioxidant, animal studies, case reports, book chapters, and conference abstracts were excluded. Our initial search yielded 395 publications based on title and abstract, while following a thorough evaluation, 335 articles were subsequently excluded based on our predefined selection criteria. In the end, 60 were considered eligible for full-text assessment. Additional articles were retrieved from a reference list hand search of the eligible studies, ending up in a total of 89 included studies (Figure 1*)*. A summary of studies based on their type can be found in Table A1 (see Appendix A).

All the included studies were subjected to an independent full-text assessment by two authors (E.N.S., I.M.). Any discrepancies during the above steps were rigorously discussed and resolved through consensus.

## 3. Redox Balance Implies Moderation-“μέτρον”

All aerobic cells are normally exposed to a degree of oxidative effects. When the levels of oxidizing agents are extremely elevated, the cells enter a state of oxidative stress (OS) [23]. “Oxidative stress” is described as the phenomenon of excessive production of active free radicals with simultaneous limited defensive action of antioxidants that ultimately leads to cell damage [24,25]. Free radicals are mainly oxygen molecules containing one or more unpaired electrons in atomic or molecular orbitals (reactive oxygen species—ROS). In addition, a subcategory of active free radicals is derived from nitrogen (reactive nitrogen species—RNS)and sulfur (reactive sulfur species-RSS) [23]. Free nitrogen radicals have been associated with oxidative effects, mediating cytotoxic and pathological conditions mainly related to inflammation [26]. Although in normal concentrations, the action of NO has been associated with physiological processes maintaining sperm motility, its sub-optimal or supra-optimal synthesis seems to cause a deterioration in sperm function, manifested as impaired count, motility, morphology, and vitality. The causative mechanisms inducing these harmful effects include the inhibition of mitochondrial respiration; ATP depletion; and induction of DNA deamination, oxidation, and nitration [27]. Reactive sulfur species have been less studied as oxidative stress inducers in the male reproductive system. H_2_S, their main precursor, has been related to normal erectile function and sperm motility at physiological levels, while its insufficiency has been implicated in impaired spermatogenesis and blood–testis barrier defects. Excessive RSS levels may also be detrimental to sperm movement owing to low energy levels caused by the inhibition of mitochondrial electron transport and ATP generation [28].

Free radicals are involved in the development of more than 100 conditions, covering a wide range from arthritis and connective tissue disorders to carcinogenesis, including aging, exposure to toxins, injury, and inflammation. Furthermore, OS has been extensively studied as one of the most important causes of infertility, both male and female [26,29,30].The increased presence of oxidants negatively affects spermatozoa and oocytes, associated with problems in gamete quality, fertilization, early embryonic development, implantation, and pregnancy rates in both natural and assisted reproduction [6,30].

Recent studies have shown that elevated ROS levels are detected in the semen of 25–40% of infertile men [31,32]. Further, 30–80% of unexplained and idiopathic male infertility cases have been causally attributed to the detrimental action of ROS. Various factors have been associated with the induction of seminal OS: lifestyle factors (e.g., smoking, alcohol consumption, obesity, psychological stress, advancing age, malnutrition), environmental exposure (e.g., high temperature, pollution, heavy metals, pesticides, phthalates), infection (genital or systemic), autoimmune (vasectomy reversal, chronic prostatitis), testicular (varicocele, cryptorchidism, torsion), and chronic disease (diabetes, kidney disease, hemoglobinopathies, hypercystinaemia, spinal cord injury) [21,23,33,34,35,36].

Immature sperm forms presenting cytoplasmic residues, round cells from various stages of spermatogenesis, and leukocytes are possible sources of active free radicals in ejaculate. Polymorphonuclear granulocytes, which make up 50–60% of all peroxidase-positive leukocytes in semen, originate mainly from the prostate and seminal vesicles. The presence of inflammation or infection activates these leukocytes, inducing the production of excessive ROS amounts [37]. In spermatozoa, the presence of cytoplasmic residues has been associated with ROS production by the mediation of the cytosolic enzyme glucose-6-phosphate-dehydrogenase (G6PD), in the cell membrane and at the mitochondrial level [25,29,38].

Spermatozoa, like all aerobic cells, are subjected to the “oxygen paradox”; that is, while the presence of free radicals at low concentrations is an important factor for cellular function (acrosome reaction, hyperactivation, capacitation, progression, and sperm-egg interaction), their action is detrimental at excessive levels [25]. All cellular macromolecules are potential targets for OS. Free radicals tend to engage in chemical reactions that remove their unpaired electron, leading to the oxidation of membrane lipids, protein amino acids, and carbohydrates within the nucleic acids [23]. Sperm cell membranes are very rich in polyunsaturated fatty acids, which are sensitive to the action of free radicals causing lipid peroxidation that negatively affects sperm motility, acrosome reaction, and fusion with the oocyte [23,25,39,40].

OS is also one of the leading causes of sperm nuclear DNA damage, occurring during passage through the epididymis [39]. Under normal conditions, sperm DNA is protected from oxidative attack in its tightly packaged form by protamines and the antioxidant activity of seminal plasma. However, infertile men often show deficient protamination, which makes the DNA molecule particularly vulnerable to oxidative effects [23]. DNA damage may be exhibited as base alterations, production of base-free genetic loci, deletions, chromosomal rearrangements, and single or double-strand DNA fragmentation [41]. Genetic mutations (point mutations or polymorphisms) have been linked to the effect of OS and the consequent induction of deterioration in sperm quality [25,29,41].

ROS can also negatively affect mitochondrial DNA (mtDNA), causing mutations mainly associated with reduced sperm motility [42,43]. The disruption of the inner and outer membranes of sperm mitochondria induces the release of cytochrome C proteins and activating caspases (detected in the anterior part of the acrosome and sperm midpiece), leading to apoptosis [25,42,43]. More importantly, rare but severe genetic diseases in the offspring, e.g., myoglobinuria and neurological deafness, have been associated with sperm mtDNA mutations caused by oxidative stress [31].

The subtle balance between oxidants and antioxidants may be disturbed by various factors that ultimately result in extreme values in both directions, causing either oxidative or reductive stress [22]. Reductive stress (RS) has been associated with adverse effects in basic seminal parameters (motility, concentration, and morphology) and fertility [44]. Excessive amounts of antioxidants may deter the fertilization process owing to the inhibition of significant functional activities of the spermatozoa [45]. In other words, over-the-counter administration of these regimens alters the redox equilibrium to a reductive-stress-state, which can even prove detrimental for reproduction, resembling the repercussion of oxidative stress (Figure 2) [46,47]. All these observations have led to the introduction of the “antioxidant paradox”, a phenomenon termed by Halliwell et al. in an attempt to describe that supplementation of large amounts of antioxidants to humans had no preventative or therapeutic effect at all [44].Ultimately, it becomes evident that the human body has to maintain a delicate oxidative-reductive environment, and “*Μέτρον or Moderation*” is a prerequisite for redox balance (Figure 3).

## 4. Antioxidant Treatment for Male Infertility—Friend or Foe?

Although empirical medical treatment with selective estrogen receptor modulators and aromatase inhibitors has already been tested for idiopathic male infertility, a more promising and cost-effective solution with antioxidant administration staggeringly gained traction, offering sperm improvement and better pregnancy outcomes [7,48,49]. The envisaged role of antioxidants is based on the premise that the administration of these nutraceuticals could easily, safely, and effectively counteract the seminal OS state [1,6].

As demonstrated in different studies, antioxidants can exert a positive impact on sperm parameters. Balercia et al. reported one of the first studies evaluating the efficacy of L-carnitine (LC), L-acetyl-carnitine (LAC), and a combination of LC and LAC treatment in improving semen kinetic parameters and the total oxyradical scavenging capacity in semen. Sixty infertile patients affected by idiopathic asthenozoospermia, with a mean age of 30 years, were enrolled into a placebo-controlled double-blind randomized trial. The supplementation with LC and LAC proved effective in increasing sperm kinetic features and improving total seminal antioxidant capacity. In addition, patients with lower baseline motility values and total oxyradical scavenging capacity were more susceptible to respond to therapy [50].

In order to determine whether the exogenous administration of coenzyme Q10 (CoQ10) was effective in improving semen quality in men with idiopathic infertility, a prospective placebo-controlled randomized study by the same group examined 60 infertile patients (age range: 27–39), comparing patients who received CoQ10 200 mg/day versus patients who received placebo over a 6-month treatment period. It was shown that patients treated with CoQ10 had more favorable sperm kinetic features than those in the placebo group after six months. More specifically, sperm cell total motility and forward motility were significantly improved in the CoQ10 treatment arm from 33.14% ± 7.12% to 39.41% ± 6.80% (*p* < 0.0001) and from 10.43% ± 3.52% to 15.11% ± 7.34% (*p* = 0.0003), respectively. On the contrary, no specific changes were noticed in the placebo group. Another significant issue in this study was that patients who underwent CoQ10 therapy were involved in twice as many spontaneous pregnancies as those in the placebo arm (6 vs. 3) during the observation period. Interestingly, patients with lower baseline motility and CoQ10 levels had a significantly greater possibility of being treatment responders. It also appeared that CoQ10 oral supplementation was overall well-tolerated, and increased concentrations of this nutraceutical may have a crucial role in the treatment of asthenozoospermic males [51].

In another study, Tremellen et al. sought to examine the effect of Menevit^®^ (Bayer, Sydney, Australia), an antioxidant consisting of vitamins C and E, selenium, garlic, lycopene, zinc, and folic acid, on embryo quality and pregnancy outcomes during in vitro fertilization-intracytoplasmic sperm injection (IVF-ICSI) treatment. For that reason, the authors conducted a prospective randomized placebo-controlled trial and attempted to test the hypothesis that Menevit^®^ would reduce oxidative sperm damage, leading to an improvement in embryo quality and pregnancy rates. Infertile males who received this compound had significantly better viable pregnancy outcomes than those who received the placebo (38.5% vs. 16%, *p* = 0.046), albeit with no significant changes in oocyte fertilization rate or embryo quality between the two groups. Of note, Menevit^®^ demonstrated a good safety profile with minimal side-effects (8%), mostly of mild nature [52].

To evaluate the efficacy of supplementation with selected natural antioxidant compounds on sperm parameters of infertile males with or without varicocele, Busetto et al. conducted a single-center prospective double-blind placebo-controlled study. They examined 104 infertile patients with oligo-astheno-teratozoospermia (mean age: 32.5 years). All participants were treated with Proxeed Plus^®^ (Sigma-Tau Health Science International B.V., Utrecht, The Netherlands), an advanced antioxidant formula consisting of 1000 mg LC, 500 mg LAC, l000 mg fructose, 725 mg fumarate, 90 mg vitamin C, 50 mg citric acid, 10 mg zinc, 20 mg CoQ10, 50 μg selenium, 200 μg folic acid, and 1.5 μg vitamin B12. Compared with the placebo group, patients who received this regimen scored markedly better on sperm parameters at the end of the trial. Sperm concentration (*p* = 0.0186), total sperm count (*p* = 0.0117), progressive motility (*p* = 0.0088), and total motility (*p* = 0.0120) were significantly increased in the supplementation treatment arm compared with the placebo group. Aside from having varicocele or not, sperm improvement remained statistically significant in all infertile patients treated with a Proxeed Plus^®^. Remarkably, 10 out of 12 pregnancies during the follow-up favored the supplementation arm, although the pregnancy rate was not a study endpoint [53].

Similarly, Micic et al. presented their initial experience with Proxeed Plus^®^ (Sigma-Tau Health Science International B.V., Utrecht, The Netherlands). They confirmed the beneficial effects of carnitine derivatives and micro-nutritive substances on sperm parameters in a prospective study involving 175 males afflicted by idiopathic oligo asthenozoospermia. Compared with the baseline values, patients who received Proxeed Plus^®^ had statistically significant increases in sperm volume, progressive motility, and vitality (*p* < 0.001) after six months of therapy. Moreover, sperm DNA fragmentation index (DFI) significantly declined compared with the values obtained before therapy (*p* < 0.001) and those obtained three months after therapy (*p* = 0.014), and decreased levels of DFI represented a reliable predictor of progressive sperm motility >10%. Simultaneous measurement of changes in sperm vitality and DFI gave the highest probability of sperm motility 10% (AUC = 0.924; 95% CI = 0.852–0.996; *p* < 0.001). Furthermore, elevated levels of seminal carnitine and α-glycosidase were positively correlated with improved progressive motility [54].

In a subsequent study by Safarinejad et al., a total of 228 males with unexplained infertility were randomly assigned 1:1 to receive orally either 200 mg of a reduced form of CoQ10 (Ubiquinol) or a comparable placebo regimen for 26 weeks. The authors primarily aimed to examine the sperm density, motility, and strict morphology within a double-blind setting. Significant increases were observed in patients who received ubiquinol compared with those receiving placebo in sperm density, motility, and morphology at the end of the treatment period (28.7 ± 4.6 × 10^6^/mL vs. 16.8 ± 4.4 × 10^6^/mL, *p* = 0.005; 35.8% ± 2.7% vs. 25.4% ± 2.1%, *p* = 0.008; 17.6% ± 4.4% vs. 14.8% ± 4.1%, *p* = 0.01, respectively). Overall, ubiquinol was deemed a safe, well-tolerated, and effective option to improve the sperm parameters, as mentioned above in males with unexplained oligo-astheno-teratozoospermia [55].

To evaluate the impact of alpha-lipoic acid (ALA) supplementation on sperm and seminal oxidative stress biomarkers, Haghighian et al. tracked 44 infertile men with idiopathic asthenozoospermia attending the infertility clinic of Ahvaz Jundishapur University of Medical Sciences in Iran. After randomization, twenty-three patients received 600 mg ALA once daily, and twenty-one patients received a matching placebo for three months. While there were no differences in the ejaculate volume and normal morphology between the two groups (*p* > 0.05), total sperm count, sperm concentration, and sperm motility were significantly higher in the ALA group at the end of the study (*p* < 0.001). Another interesting point is the fact that ALA supplementation demonstrated the potential to significantly improve total antioxidant capacity (TAC) (1.13 ± 0.42 vs. 1.78 ± 0.40 μmol/L; *p* = 0.001) and malondialdehyde (MDA) levels (*p* = 0.002) compared with the placebo arm [56].

In 2015, Calogero et al. prospectively tracked 194 patients with idiopathic infertility who did not achieve pregnancy for more than two years of unprotected sexual intercourse. The patients were subsequently randomized and treated with myoinositol (MYO) in order to evaluate the effects of this compound on sperm function and reproductive hormones. A total of 98 patients received myoinositol (MYO) 4 gr/day, and 96 patients received placebo treatment for three months. Compared with the placebo arm, patients receiving MYO had a significantly increased percentage of acrosome-reacted spermatozoa, sperm concentration, total count, and progressive motility. Accordingly, MYO rebalanced serum luteinizing hormone, follicle-stimulating hormone, and inhibin B concentrations without causing adverse reactions. Astonishingly, to the best of the author’s knowledge, this study was the first to assess MYO in the treatment of idiopathic infertility [57]. Many studies also examined the MYO’s role in fertility enhancement either in vitro or in vivo [58,59,60,61].

Generally, the rationale behind the administration of antioxidants is that they can improve the total antioxidant buffering capacity of seminal plasma while simultaneously reducing the levels of seminal ROS with the least possible treatment-related adverse events [19].

Despite the above-mentioned beneficial effects, evidence-based data are still lacking, generating controversy and indicating that antioxidant therapy has pros as well as cons [62].

Firstly, Rolf et al., from the Institute of Reproductive Medicine of Münster, attempted to investigate the hypothesis that a combination of antioxidant molecules would exert a dose-dependent effect on sperm parameters and conducted a double-blind, randomized placebo-controlled trial enrolling thirty-three infertile males. Unfortunately, no specific improvement of conventional sperm parameters, 24 h sperm survival rate, or the initiation of pregnancies could be demonstrated after completing the combined high-dose antioxidant treatment (1000 mg vitamin C + 800 mg vitamin E) [63].

Similar to Rolf et al., Greco et al. studied sixty-four men with unexplained infertility who were treated with a comparable combined antioxidant regimen consisting of 1 g vitamin C and 1 g vitamin E. Two-month therapy failed to significantly improve sperm concentration, count, motility, and morphology in either the treatment or the placebo arm [64].

Furthermore, Silver et al. analyzed the impact of vitamin C, vitamin E, and beta-carotene on sperm chromatin integrity of eighty-seven healthy male volunteers. It was found that men with moderate, but not high beta-carotene intake had an increase in standard deviation DNA fragmentation index (DFI) compared with participants with a low intake (adjusted means 206.7 and 180.5, respectively; *p* = 0.03), as well as an increase in the percentage of immature sperm (adjusted means 6.9% and 5.0%, respectively; *p* = 0.04) [65].

Another significant issue concerned the daily use of oral carnitine (2000 mg LC + 1000 mg LAC) in ameliorating the semen parameters of men with idiopathic asthenospermia. Among the twenty-one patients who entered the double-blind placebo-controlled study, no patient faced a statistically or clinically significant increase in sperm motility or total motile sperm counts at sequential treatment time points (12-week, 24-week) in both carnitine and placebo arms [66].

Increasing evidence suggests that irrational administration of antioxidants may lead to the disruption of the balance between oxidative and reductive state with immediate effect on sperm physiology. Ménézo et al. conducted a bicentric study enrolling patients who had at least two previous failures of IVF or ICSI. They found that patients who were administered vitamins C and E (400 mg each), β-carotene (18 mg), zinc (500 μmol), and selenium (1 μmol), for 90 days, had a significant improvement in DFI (−19.1%, *p* < 0.0004), albeit with a bitter tradeoff of a significant increase in sperm decondensation (+22.9%, *p* < 0.0009). The authors have sounded the alarm about a logical antioxidant administration, especially in patients with a degree of decondensation over a threshold of 20%, to avoid reaching the critical value of 28%. They have also partly explained this unexpected paradoxical effect owing to the opening of interchain disulfide bridges in protamines with subsequent interference in the paternal gene activity during preimplantation development [67]. Similarly, the same author in another study underlined the significance of the interracial variability, which leads to different proclivity among human races for isolated nutraceutical deficiencies (i.e., zinc, or selenium) [68]. Likewise, overdosing of selenium may alter the sperm selenoproiten(s) of the outer mitochondrial membrane and alter the thyroid hormone metabolism, both conditions associated with asthenospermia [69,70]. The irrational use of beta-carotene, a precursor (inactive form) to vitamin A, may also cause a side effect to sperm physiology. Murata et al. showed that the autoxidation of retinoids produce superoxide, dismutated to H_2_O_2_, which ultimately leads to DNA damage in the presence of endogenous metals [71].

Recently, a multicentric double-blind, randomized placebo-controlled trial from the United States of America examined 85 patients who were treated daily with an antioxidant formulation containing vitamin C (500 mg), vitamin E (400 mg), selenium (0.20 mg), L- carnitine (1000 mg), zinc (20 mg), folic acid (1000 mcg), and lycopene (10 mg), and 86 males who received placebo. While sperm concentration differed between the antioxidant group (−4.0 (−12.0, 5.7) M/mL) and the placebo group (+2.4 (−9.0, 15.5) M/mL) (*p* = 0.03), no significant differences between the two groups in change in sperm morphology, motility, or DNA fragmentation could be demonstrated after three months of treatment. DNA fragmentation did not differ among the 44 men with high DNA fragmentation (29.5 (21.6, 36.5) % versus 28.0 (20.6, 36.4) %; *p* = 0.58) at 3 months. Interestingly, cumulative live birth did not differ between the antioxidant and placebo groups (15% versus 24%; *p* = 0.14) at six months. This study was mitigated by the limited sample size; nonetheless, these findings suggested that administration of antioxidants in the male partner of the couple did not improve in vivo pregnancy or live birth rates, and no improvements in semen parameters or DNA integrity could be observed either [72].

Without doubt, irrational administration gives birth to the well-established antioxidant paradox, a paradoxical phenomenon suggested by Halliwell, rendering the whole supplementation procedure a critical challenge [6].

The administration of antioxidants has constantly fueled a debate about whether this approach should be considered logical for treating infertile males. In addition, conflicting data from the above trials prompted various groups to systematically analyze evidence relevant to the role of these compounds in counteracting male infertility.

Recently, Salvio et al. performed a systematic review to evaluate the usefulness of CoQ10 in treating male infertility, either alone or adjunctive to other molecules. It was shown that supplementation with CoQ10 significantly increased sperm quality, and especially sperm motility. A favorable effect was also seen in sperm concentration, while the effects on normal morphology appeared to be lower. Most improvements were noticed after 3–6 months of treatment initiation, but gradually faded following administration withdrawal. Considering that the exact dosage of this compound remains unknown to date, the authors concluded that CoQ10 therapy should be regarded complementary to other therapies and not a standalone alternative option. Meanwhile, in most studies, a dosage of 200 mg/day was viewed as the optimal dosage to achieve favorable results [73].

Lafuente et al. had previously confirmed the advantageous effect of CoQ10 treatment in a meta-analysis of three randomized placebo-controlled clinical trials incorporating 149 patients in the CoQ10 group and 147 in the placebo group. None of the included studies provided sufficient details relevant to live births, albeit this was intended to be a primary outcome. It was shown that patients who were treated with CoQ10 demonstrated a statistically significant increase in CoQ10 seminal concentration (RR 49.55, 95% CI 46.44 to 52.66, I(2) = 17%), sperm concentration (RR 5.33, 95% CI 4.18 to 6.47, I(2) = 58%), and sperm motility (RR 4.50, 95% CI 3.92 to 5.08, I(2) = 0%). It also appeared that CoQ10 was not associated with an increase in pregnancy rates. The authors underlined that more detailed research would further clarify the exact role of CoQ10 in clinical practice [74].

Moreover, a promising new molecule, alpha-lipoic acid (ALA), in the field of couple infertility was recently examined within a systematic review, demonstrating favorable outcomes for sub-fertile women and for the sperm quality of infertile males [75].

In a Cochrane systematic review of 61 studies, 6264 sub-fertile males (age range: 18–65) were treated with 18 different oral antioxidants and faced encouraging results in live birth and clinical pregnancy rates, with miscarriage rates remaining very low. Considering the low quality of evidence, it was shown that antioxidants may lead to increased live birth rates (OR 1.79, 95% CI 1.20 to 2.67, *p* = 0.005, 7 RCTs, 750 men, I^2^ = 40%) and a corresponding increase in clinical pregnancy rates (OR 2.97, 95% CI 1.91 to 4.63, *p* < 0.0001, 11 RCTs, 786 men, I^2^ = 0%). Overall, the evidence was deemed inconclusive, and further large-scale, well-designed randomized placebo-controlled trials reporting on pregnancy and live births would be more than essential to clarify the exact role of antioxidants [76].

A system review by Ross et al., examining seventeen randomized trials, including a total of 1665 males, supported the notion that the use of antioxidants is justified in male infertility, offering a potential improvement in sperm quality and pregnancy rates. More specifically, ten trials examined pregnancy rates, and six demonstrated improvement after antioxidant therapy [77]. Similarly, Majzoub et al. corroborated the previous findings in a systematic review of 26 studies, demonstrating the positive effect of antioxidants on basic semen parameters, advanced sperm function, outcomes of assisted reproductive therapy, and livebirth rate [78].

Agarwal et al. reported comparable beneficial effects on sperm and pregnancy outcomes in a systematic review of 97 well-designed studies, with approximately half of them (46%) being RCTs and 45.4% testing individual antioxidant regimens. The authors noteworthily gave an illustrative overview of the impact of antioxidant supplementation in infertility treatment based on a strength weakness opportunity threat (SWOT) analysis [62]. Previously, Esteves utilized an analogous SWOT analysis to underline the perceived advantages and drawbacks of SDF as a specialized sperm function test in clinical practice [79].

The issue of whether there is a correlation between seminal plasma zinc concentrations and male infertility and the effects of this essential trace mineral on sperm parameters has been the subject of a thorough investigation by Zhao et al., who performed a meta-analysis of twenty studies. The authors found that zinc significantly increased semen volume, sperm motility, and the percentage of normal sperm morphology (standard mean differences, SMD) [95% CI]: −0.99 [−1.60, −0.38], −1.82 [−2.63, −1.01], and −0.75 [−1.37, −0.14], respectively. Additionally, it was shown that the concentration of zinc in the seminal plasma was significantly lower than those from the normal controls (SMD) [95% CI] −0.64 [−1.01, −0.28] [80].

In 2020, Tsampoukas et al. investigated the role of LC as a primary or complementary treatment in infertile patients with varicocele. Their systematic review, including four studies, resulted in equivocal results, thus posing dilemmas related to the definitive recommendation of this regimen in this particular population [81]. One year later, the same group attempted to systematically assess whether vitamins, either in the form of antioxidant complex or solely administered, could be used as primary or adjuvant therapy for infertile males with varicocele. Although the results from the qualitative analysis of seven studies appeared promising, future research should evaluate the pregnancy rates as a primary endpoint [82].

Contrary to the studies mentioned above, our group, in a recent systematic review and meta-analysis of 14 studies, addressed that antioxidant supplementation does not seem to improve the pregnancy rate, semen parameters, or DNA integrity in patients with varicocele-associated infertility. As far as pregnancy rates are concerned, no significant differences were demonstrated after antioxidant treatment versus no treatment at three (OR: 2.28, 95% CI: 0.7–7.48) and six months (OR: 1.88, 95% CI: 0.62–5.72) in patients who underwent varicocele surgical repair. Given the controversial finding for sperm concentration, morphology, motility, and DNA fragmentation, no definite conclusions regarding the optimal type, dosage, and treatment duration could be safely drawn [83].

What is noticeable is that the number of patients embracing these natural therapies as safe and easily accessible is steadily growing [6]. Taking this one step further, we recognize a wide selection of antioxidants from which to choose, either alone or in a combination with micro-nutrients, most of which are readily available at a low cost [84].

While many studies have already highlighted the advantageous effects of antioxidants on sperm parameters, there is much more to be clarified in the ever-changing field of male infertility [7,22]. Moreover, the ability to translate these sperm changes into improved chances of pregnancy is less clear to date. Hence, more adequately powered placebo-controlled trials with well-defined outcomes are warranted and will help further standardize the generalizability of this approach [22,85].

On the one hand, antioxidants should not be considered as a foe, as long as the risks are weighed against the benefits. On the other hand, uncontrolled supplementation always carries risks falling into a paradoxical effect, thus crossing the thin dividing line between hope and threat from their use [44]. An overview of studies reporting on the beneficial and detrimental effects of antioxidants can be found in Table 1.

## 5. Oxidation-Reduction Potential (ORP)—A Reliable Guarantor for Redox Balance?

Varicocele, cigarette smoking, diabetes, stress, alcohol consumption, aging, and ionizing radiation are partly or wholly responsible for OS exacerbation. This is mostly reflected through the abnormal levels of oxidation-reduction potential (ORP). ORP represents a balance between oxidants and antioxidants (reductants), a fundamental measure of the redox system, especially OS [12]. Generally, it could be advocated that ORP measurement is a meaningful step towards a comprehensive male infertility assessment. As OS upholds a state of redox regulation where oxidants overwhelm the antioxidant’s scavenging capacity, a diagnostic workup encompassing ORP evaluation would be a reliable way to analyze semen’s quality [12].

Oxidative stress is highly implicated in the pathophysiology of idiopathic male infertility, and undoubtedly, the conventional semen analysis may not suffice to adequately assess vital aspects of the fertilization process. Therefore, a rapid in vitro diagnostic test offering ORP quantification would be of great value, further minimizing impractical referrals to fertility clinical laboratories [7].

Although numerous diagnostic tools have been previously tested for direct and indirect OS measurement, no test could predominate and provide a composite measurement of known and unknown oxidants and antioxidants (Table 2) [19]. Currently used assays, namely, malondialdehyde assay (MDA), total antioxidant capacity (TAC), chemiluminescence for ROS, and ROS-TAC score, have been judged for their burdensome cost, methodological complexity, and need for large sample volumes [19,35]. Moreover, they are considered tedious and time-consuming, making any incorporation into clinical practice a critical challenge [18].

In this setting, MiOXSYS^®^, a galvanostat-based analyzer, emerged as a clinically useful diagnostic option conferring various advantages, primarily providing a complete snapshot of a male’s OS state [20]. It virtually requires only 30 μLof a sample, taking less than 4 min to complete the whole test. It undeniably offers a reliable, reproducible, and practical way to assess sperm quality with high sensitivity and specificity and practical advantages over the previously reported diagnostic assays [7].

Urologists take advantage of this quick, easy-to-use, and accurate ancillary to the basic semen analysis tool, which consistently predicts an individual’s fertilization capacity via monitoring OS alterations to a pro-oxidant state. With demonstrable results correlating ORP levels with sperm concentration, count, and motility in infertile patients, it is expected to guide future therapeutic interventions for males with MOSI [7]. Ultimately, ORP is significantly negatively correlated with concentration, motility, and count, whereas it is strongly positively correlated with SDF in infertile males [18].

Recently, Agarwal et al. proposed that testing of ORP levels in MOSI patients treated with antioxidants would be of great importance, allowing for compliance confirmation and treatment monitoring [7]. They have also underlined the hesitancy of clinicians in routinely measuring OS as part of the diagnostic workup owing to the absence of standardized protocols. In the meantime, an ORP cut-off value of 1.34 mV/10^6^ was suggested as reliable for detecting abnormal/normal semen quality [18].

Trying not to forget the Hippocratic albatross, “ὠφελέειν, ἢ μὴ βλάπτειν/first do no harm”, we acknowledge that ORP measurement via MiOXSYS^®^ plays a key role in guaranteeing redox balance in male infertility.

## 6. Conclusions, Future Directions, and Perspectives

In this review, we explored the vicious cycle of OS; we emphasized the ultimate need for moderation or “*Μέτρον*” in male infertility and summarized critical data on clinical studies supporting the use of antioxidants as a defense shield against OS. Finally, we provided a glimpse of reductive stress and the antioxidant paradox, and elucidated the promising concept of ORP measurement by the novel MiOXSYS method.

Male infertility, a prevalent condition afflicting 15% of couples globally, remains undoubtedly a burdensome health problem demanding specific answers [3,86]. OS has been shown to be highly implicated in its pathogenesis, with elevated ROS levels inducing sperm dysfunction [19,87]. Motility, morphology, total count, and viability alterations are among the most significant repercussions of OS on sperm’s structural and functional integrity. OS has also been incriminated for DNA strand breaks and chromatin cross-linking, resulting in SDF [11]. Small circulating, physiological amounts of ROS are essential for reproduction and the processes of capacitation, hyperactivation, acrosome reaction, and sperm–oocyte binding [8].

In this context, we conveyed the message that redox balance is crucial and this conforms with the dictum “*Μέτρον ἄριστον*”, asserting excellence in moderation and implying that extreme diversions in either side are unlikely to be the best practice. Any disruption in cellular redox homeo-dynamics has a profound impact on sperm function and fertilization [8].

Regarding the observed trends towards empiric antioxidant administration, all we shall ever learn is that irregular use poses a threat to redox homeostasis [6]. Lately, an unprecedented surge in antioxidants’ administration has been noted owing to the ease of their use and wide availability [7]. The fervent interest in their broad prescription is partly justified by the growing body of evidence supporting their effectiveness in ameliorating sperm parameters [78,85].

In general, we could argue that this approach appears oversimplified, and controversies regarding their beneficial use still exist [6]. Robust prospective placebo-controlled trials have to deal with their efficacy in improving live birth rates [7]. Physicians must bear in mind the antioxidant paradox because of the general propensity to widely prescribe these regimens [44]. Of note, Aitken aptly paralleled the infertile patient with the one presenting in a hospital with a coma. Insulin administration would only be beneficial if glycemic aberration is the underlying cause of the problem. In this setting, only those in need of insulin would get better and recover. Conversely, if insulin is arbitrarily given to coma patients irrespective of their glycemic status, this could have a disastrous effect on the patient’s health [16]. Hence, it becomes evident that antioxidants are actually justified only in those patients exhibiting signs of OS; otherwise, their use exacerbates sperm cell injury, leading to a reductive stress state [7].

Interestingly, there are many unanswered questions on the optimal type of antioxidant, exact dosage, treatment intervals, and total duration of therapy [6]. More importantly, no clear-cut recommendations in the form of guidelines have been implemented into clinical practice, a fact that raises skepticism over their broad use. Thus, there is an imperative need for studies that will clarify the issues mentioned earlier, further assessing antioxidants’ long-term impact on male infertility [6].

Over the last century, conventional semen analysis has been the surrogate marker of a male’s fertility potential. Nevertheless, it would be no exaggeration to admit that this diagnostic method has totally reached the limit [19]. For instance, relying only on basic sperm analysis, a vast majority of patients presenting with idiopathic male infertility would have been diagnostically underprivileged. It becomes apparent that a single analysis is not enough for investigating a male’s fecundity [12]. Future research should incorporate more aspects of a male’s fertilization capacity with a meaningful evaluation of OS or SDF. Remarkably, there are no guidelines to incorporate a broad examination of ROS into everyday clinical practice [7].

Despite the plethora of tedious and time-consuming assays assessing OS, a novel user-friendly technique has been developed with the ability to measure both oxidants and antioxidants at one time [18]. The advent of MiOXSYS is anticipated to push the field forward and adjust critical parameters relevant to OS regulation. MiOXSYS worthily represents “the new kid on the block”, demonstrating the full potential to supplant previously used OS measurement assays. Furthermore, its vital role as a gold standard ancillary diagnostic tool in MOSI remains to be substantially validated in the long run. Whether it will be the guarantor of redox balance, securing “*Μέτρον*”, is a question whose answer will be addressed upon in-depth research.

As urologists continue to face the vexing dilemma of administering antioxidants, it becomes clear that developing a diagnostic workup algorithm and new evidence-based treatment guidelines should constitute a dynamic and ongoing process on the horizon [62]. Furthermore, the separation between MOSI-positive and MOSI-negative males with idiopathic or unexplained infertility will improve screening standardization and identify any potential modifiable factors [88]. Therefore, developing validated OS measurement tools like MiOXSYS will help monitor and guide therapeutic interventions while driving strategic decision-making in infertile couples [7].

Lastly, more elaborate studies are expected to shed light on specific cellular signaling pathways based on ORP measurement, thus offering a better understanding of the etiology of MOSI [7,89]. Without doubt, every approach to the infertile male should endorse redox balance maintenance [6,44]. MiOXSYS appears to be the key starting point to fulfill the need to test for OS. Well-designed, large-scale prospective trials are indeed warranted and will allow for robust scientific knowledge acquisition [5]. The future of male infertility seems definitely bright, and a lot more is yet to come.

## Figures and Tables

**Figure 1 antioxidants-10-01534-f001:**
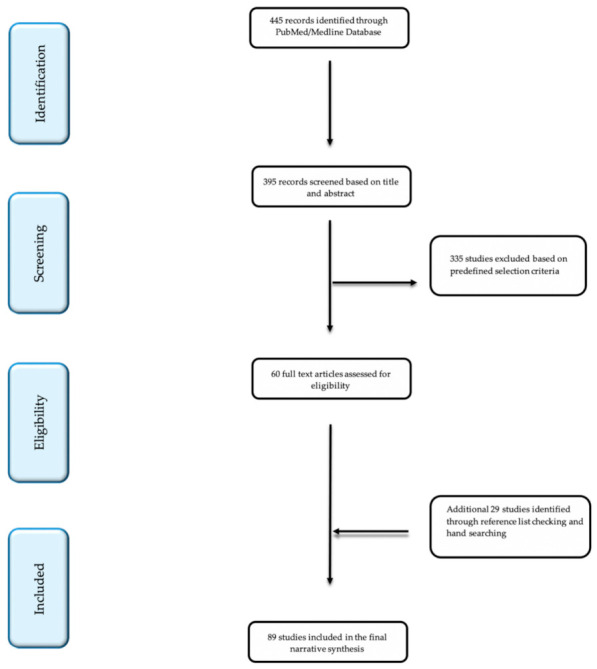
Flow diagram of studies included in the narrative synthesis.

**Figure 2 antioxidants-10-01534-f002:**
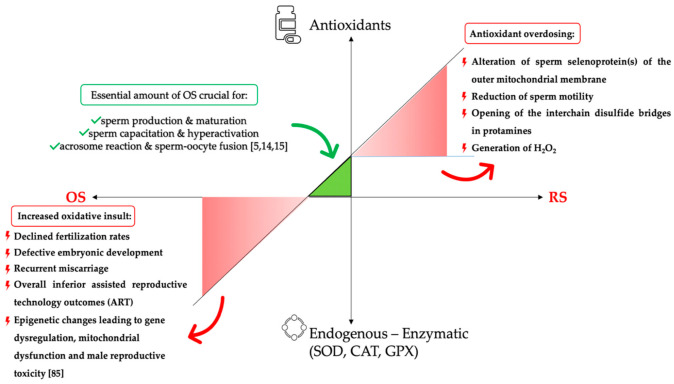
Schematic presentation of the antioxidant administration. The green triangle shows the area of the essential amount of OS necessary for the physiological function of the spermatozoa. Depletion of antioxidant reserves or increased oxidative insult results in OS (red triangle—down-left), whereas irrational use of antioxidants will result in RS (red triangle—up-right). CAT: catalase; GPX: glutathione peroxidase; H_2_O_2_: hydrogen peroxide; OS: oxidative stress; RS: reductive stress; SOD: superoxide dismutase.

**Figure 3 antioxidants-10-01534-f003:**
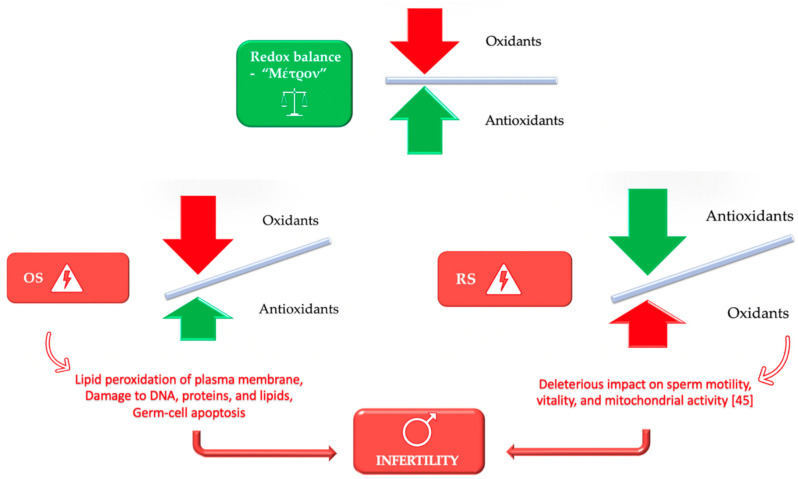
Redox balance in male infertility and the need for moderation-“Μέτρον”.

**Table 1 antioxidants-10-01534-t001:** Summary of studies reporting on the beneficial and detrimental effects of antioxidants. Green color: beneficial effect; orange color: no specific impact or effect or no improvement; red color: detrimental effect. CoQ10: coenzyme; DFI: DNA fragmentation index; LC: L-carnitine; LAC: L-acetyl-carnitine; Se: selenium.

Author, Year	Examined Antioxidant Compound	Results
Balercia et al., 2005 [50]	LC, LAC, and a combination of LC and LAC	Sperm kinetic parameters improvement
		Total seminal antioxidant capacity increase and positive correlation with kinetic features improvement
Balercia et al., 2009 [51]	CoQ10	Sperm cell total motility and forward motility improvement
		Twice as many spontaneous pregnancies during the observation period
Busetto et al., 2018 [53]	LC, LAC, fructose, fumarate, vitamin C, citric acid, zinc, CoQ10, Se, folic acid, and vitamin B12	Significant increase in sperm concentration, total sperm count, progressive, and total motility
		10 out of 12 pregnancies during the follow-up favored the supplementation arm, although the pregnancy rate was not a study endpoint
Micic et al., 2019 [54]	LC, LAC, fructose, fumarate, vitamin C, citric acid, zinc, CoQ10, selenium, folic acid, and vitamin B12	Significant increases in sperm volume, progressive motility, and vitality after six months of therapy.
		Significant decline in sperm DNA fragmentation index (DFI) compared with baseline and 3-month therapy
		Decreased levels of DFI represented a reliable predictor of progressive sperm motility >10%
		Positive correlation of elevated levels of seminal carnitine and α-glycosidase with improved progressive motility
Safarinejad et al., 2012 [55]	Ubiquinol—Reduced form of CoQ10	Significant increases in sperm density, motility, and morphology
Haghighian et al., 2015 [56]	Alpha-lipoic acid	Significant increases in total sperm count, sperm concentration, and sperm motility
		Potential to significantly improve total antioxidant capacity (TAC) and malondialdehyde (MDA) levels
Calogero et al., 2015 [57]	Myoinositol	Significant increase in the percentage of acrosome-reacted spermatozoa, sperm concentration, total count, and progressive motility
		Rebalance in serum luteinizing hormone, follicle-stimulating hormone, and inhibin B concentrations
Tremellen et al., 2007 [52]	Vitamins C and E, Se, garlic, lycopene, zinc, and folic acid	Significant improvement in viable pregnancy rates
		No significant changes in oocyte fertilisation rate or embryo quality
Steiner et al., 2020 [72]	Vitamin C, vitamin E, Se, LC, zinc, folic acid, and lycopene	No significant improvements on sperm morphology, motility, or DNA fragmentation
		No improvement on in vivo pregnancy or live birth rates
Sigman et al., 2006 [66]	LC and LAC	No statistically or clinically significant increase in sperm motility and total motile sperm counts at sequential treatment time points
Rolf et al., 1999 [63]	Vitamin C and E	No specific improvement on conventional sperm parameters, 24 h sperm survival rate, or pregnancy initiation
Greco et al. 2005 [64]	Vitamin C and E	No significant improvement on sperm concentration, count, motility, and morphology
		Significant decline in DNA-fragmented spermatozoa
Silver et al. 2005 [65]	Vitamin C, vitamin E, and beta-carotene	Moderate, but not high beta-carotene intake increased standard deviation of DNA fragmentation index (SD DFI), as well as the percentage of immature sperm, compared with low intake
Hawkes et al., 2001 [70]	Se	Overdosing of Se may alter the sperm selenoprotein(s) of the outer mitochondrial membrane and alter the thyroid hormone metabolism both conditions associated with asthenospermia and impaired fertility
Ménézo et al., 2007 [67]	Vitamins C and E, β-carotene, zinc, and Se	Decrease in sperm DNA fragmentation
		Increase in sperm decondensation
Bleau et al. 1984 [69]	Se	Se level between approximately 40 and 70 ng/mL was optimal for reproductive performance (high pregnancy rate and low abortion rate)
		Sperm motility was maximal at semen Se levels ranging between 50 and 69 ng/mL
		Se levels ≤35 ng/mL were associated with male infertility
		Se levels ≥80 ng/mL were associated with a high abortion rate and signs of ovarian dysfunction in the partner

**Table 2 antioxidants-10-01534-t002:** Various techniques of seminal OS measurement. ROS: reactive oxygen species.

Direct and Indirect Techniques to Measure OS in the Seminal Plasma	
Direct	Indirect
Oxidation-reduction potential (ORP)	Total antioxidant capacity (TAC)
ROS by chemiluminescence	ROS-TAC
Nitro blue tetrazolium (NBT)	Malondialdehyde (MDA)
Cytochrome C reduction test	Myeloperoxidase or Endtz test
Fluorescein probe	Lipid peroxidation levels
Oxidation-induced fluorochrome probe	Chemokines
Electron spin resonance	Ascorbate
	DNA fragmentation

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
