# Peer review of "Redox Balance in Male Infertility: Excellence through Moderation—“Μέτρον ἄριστον”"

_antioxidants, 2021, doi:10.3390/antiox10101534_

Round 1

Reviewer 1 Report

General comments

In the review manuscript “Redox balance in male infertility: “Μέτρον ἄριστον” the authors offered a summary of evidence relevant to redox regulation in male reproduction and analysed the impact of OS and reductive stress on sperm function. They also examined the most up-to-date scientific literature regarding ORP and its measurement.

These topics are of great relevance and this manuscript could provide important reference information that are of considerable interest.

The manuscript is well written and the search strategy appears correct and very clear. The authors have consulted more than 400 publications and about 80 were included in the full text assessment.

Specific comments

I recommend to significantly improve the schemes proposed in the figures, both in graphics and in content.

Author Response

We would like to thank Reviewer 1 for his kind and contructive comments.

“I recommend to significantly improve the schemes proposed in the figures, both in graphics and in content.”

Authors’ answer:

The authors have significantly increased the content and the graphics of all the figures (Figure 1, Figure 2, and Figure 3) of the manuscript.

Sincerely,

Fotios Dimitriadis, MD, PhD, FEBU

Asst. Professor of Urology

Urology Department

Arisotle University of Thessaloniki

Thessaloniki, Greece

T: +302310992525

M: +306945290210

E: helabio@yahoo.gr

Reviewer 2 Report

In this manuscript the authors aim to review information on redox balance and sperm quality/human male infertility, reviewing also the possible benefits of antioxidant supplementation. Overall this is an interesting topic, although I do have issues with the text, as currently presented.

-The main objection I have is that this is presented as a review article, but includes original, unpublished data, to help make its point (line 259 and subsequent). This original data has not been peer reviewed, and cannot be as currently presented as most information I would require from this type of paper is not present. There are several options:

1- This data is fully included and the paper thus becomes more of a research paper.

2- This mention/information is removed from the article altogether (not even cited as data not shown or unpublished, completely removed).

3- The data is published and only then this review is presented, citing the publication.

Other comments:

-A Figure/flowchart including the selection (inclusion/exclusion criteria) of articles for this analysis should be included. Furthermore, a Supplementary table with all the relevant references used and a short description (original paper, review, hand picked, etc) should also be part of the manuscript.

- For completeness RSS should also be mentioned, besides ROS and RNS.

- While the Reviewer greatly appreciates the profoundness of concepts written in Greek, perhaps an English version should also be mentioned in the title and throughout the text.

-For the benefit of the reader the authors could summarize the results of the different treatment studies mentioned in section 4 in a Table.

Author Response

We would like to thank Reviewer 2 for his kind and contructive comments. We adopted all the comments and more specificaly:

The main objection I have is that this is presented as a review article, but includes original, unpublished data, to help make its point (line 259 and subsequent). This original data has not been peer reviewed and cannot be as currently presented as most information I would require from this type of paper is not present

Authors’ answer:

The authors have ommited all the unpublished data as the expert reviewer suggeted.

A Figure/flowchart including the selection (inclusion/exclusion criteria) of articles for this analysis should be included.

Authors’ answer:

The authors have added a flowchart figure (Figure 1) of the articles used in the manuscript for analysis.

Furthermore, a Supplementary table with all the relevant references used and a short description (original paper, review, handpicked, etc) should also be part of the manuscript.

Authors’ answer:

The authors have added a supplementary table (Table 1) with all the relevant references used and a short description as the expert reviewer suggeted.

“For completeness RSS should also be mentioned, besides ROS and RNS.”

Authors’ answer:

A paragraph mentioning more extensively RSS and RNS has been added as the expert reviewer suggeted (Line 167-180).

While the Reviewer greatly appreciates the profoundness of concepts written in Greek, perhaps an English version should also be mentioned in the title and throughout the text.”

Authors’ answer:

An English version has been added in the title and throughout the text as suggeted by the expert reviewer (Lines 2, and 110).

For the benefit of the reader the authors could summarize the results of the different treatment studies mentioned in section 4 in a Table

Authors’ answer:

The authors have summarized the results of the different treatment studies mentioned in section 4 in a table (Table 2) as suggeted by the expert reviewer.

Thank you very much.

Sincerely,

Fotios Dimitriadis, MD, PhD, FEBU

Asst. Professor of Urology

Urology Department

Arisotle University of Thessaloniki

Thessaloniki, Greece

T: +302310992525

M: +306945290210

E: helabio@yahoo.gr

Round 2

Reviewer 2 Report

The authors have adequately addressed my comments. I have no further concerns.